# The impact of a demand-side sanitation and hygiene promotion intervention on sustained behavior change and health in Amhara, Ethiopia: A cluster-randomized trial

**Matthew C. Freeman**[1]*, **Maryann G. Delea**[1], **Jedidiah S. Snyder**[1], **Joshua V. Garn**[2], **Mulusew Belew**[3], **Bethany A. Caruso**[4], **Thomas F. Clasen**[1], **Gloria D. Sclar**[1], **Yihenew Tesfaye**[5], **Mulat Woreta**[3], **Kassahun Zewudie**[3], **Abebe Gebremariam Gobezayehu**[3,6]

**1** Gangarosa Department of Environmental Health, Rollins School of Public Health, Emory University, Atlanta, Georgia, United States of America, **2** School of Community Health Sciences, University of Nevada, Reno, Nevada, United States of America, **3** Emory Ethiopia, Bahir Dar and Addis Ababa, Ethiopia, **4** Hubert Department of Global Health, Rollins School of Public Health, Emory University, Atlanta, Georgia, United States of America, **5** Department of Social Anthropology, Bahir Dar University, Bahir Dar, Ethiopia, **6** School of Nursing, Emory University, Atlanta, Georgia, United States of America

* matthew.freeman@emory.edu

**Data Availability Statement:** Data from the trial is available via 10.17605/OSF.IO/VH7YS. Additional

## Abstract

Behaviors related to water, sanitation, and hygiene (WASH) are key drivers of infectious disease transmission, and experiences of WASH are potential influencers of mental well-being. Important knowledge gaps exist related to the content and delivery of effective WASH programs and their associated health impacts, particularly within the contexts of government programs implemented at scale. We developed and tested a demand-side intervention called *Andilaye*, which aimed to change behaviors related to sanitation, personal hygiene, and household environmental sanitation. This theory-informed intervention was delivered through the existing Ethiopian Health Extension Programme (HEP). It was a multi-level intervention with a catalyzing event at the community level and behavior change activities at group and household levels. We randomly selected and assigned 50 *kebeles* (sub-districts) from three *woredas* (districts), half to receive the *Andilaye* intervention, and half the standard of care sanitation and hygiene programming (i.e., community-led total sanitation and hygiene [CLTSH]). We collected data on WASH access, behavioral outcomes, and mental well-being. A total of 1,589 households were enrolled into the study at baseline; 1,472 households (94%) participated in an endline assessment two years after baseline, and approximately 14 months after the initiation of a multi-level intervention. The intervention did not improve construction of latrines (prevalence ratio [PR]: 0.99; 95% CI: 0.82, 1.21) or handwashing stations with water (PR: 0.96; 95% CI: 0.72, 1.26), or the removal of animal feces from the compound (PR: 1.10; 95% CI: 0.95, 1.28). Nor did it impact anxiety (PR: 0.90; 95% CI: 0.72, 1.11), depression (PR: 0.83; 95% CI: 0.64, 1.07), emotional distress (PR: 0.86; 95% CI: 0.67, 1.09) or well-being (PR: 0.90; 95% CI: 0.74, 1.10) scores. We report limited impact of the intervention, as delivered, on changes in behavior and mental well-being. The effectiveness of the intervention was limited by poor intervention fidelity.

information is available via OSF site https://osf.io/vh7ys/.

**Funding:** This research was funded by grants to Emory University from the World Bank Group (7175829 / PI:MCF), Children's Investment Fund Foundation (1606-01334 / PI:MCF), and the International Initiative for Impact Evaluation (TW11.1016 / PI:MCF). The funders have otherwise played no role in the design of the study; collection, analysis, or interpretation of the data; or writing of the manuscript.

**Competing interests:** The authors declare no competing interests.

While sanitation and hygiene improvements have been documented in Ethiopia, behavioral slippage, or regression to unimproved practices, in communities previously declared open defecation free is widespread. Evidence from this trial may help address knowledge gaps related to challenges associated with scalable alternatives to CLTSH and inform sanitation and hygiene programming and policy in Ethiopia and beyond.

**Trial registration**: This trial was registered with clinicaltrials.gov (NCT03075436) on March 9, 2017.

## Introduction

Inadequate water, sanitation, and hygiene (WASH) are key drivers of infectious disease transmission [1, 2]. Diarrhea accounts for an estimated 1.65 million deaths annually [3] and nearly 10% of all under-5 deaths in low-income settings [4]. Deficiencies in WASH are also a major contributor of neglected tropical diseases (NTDs) [5, 6]. Over one billion people are at risk of soil-transmitted helminthiasis, which leads to nearly five million disability adjusted life years (DALYs), and schistosomiasis leads to two million DALYs [7, 8]. Trachoma, the leading infectious cause of blindness [9], is precipitated by repeat infections with *Chlamydia trachomatis* bacteria, which are often perpetuated by poor hygiene [10]. These infections are environmentally mediated [11], and are largely attributed to inadequate WASH [12, 13].

While WASH studies have primarily focused on infectious diseases or anthropometric measures of growth amongst young children, this narrow focus does not fully encapsulate the World Health Organization (WHO)'s definition of health as "a state of complete physical, mental, and social well-being and not merely the absence of disease or infirmity" [14]. A growing body of research has identified linkages between water and sanitation and mental health outcomes [14–17]. For example, extensive qualitative and quantitative research has demonstrated how water insecurity can influence mental health, particularly among women [18–21]. Research on sanitation and mental health is emergent, and predominantly qualitative [22–24]. A cross-sectional study in Odisha, India, found women's sanitation insecurity—their negative sanitation experiences and concerns—to be associated with stress, depression, distress, and impaired general well-being, even among those with access to a sanitation facility [25]. Further, a systematic review of sanitation and well-being found open defecation and use of sanitation facilities can negatively influence mental and social well-being for women and girls, especially when they experience or perceive a lack of privacy and safety [14]. As such, improvements in women's mental health likely requires more than physical access to sanitation facilities, but also gender-sensitive modifications to facilities and shifts in gender norms to improve women's experiences of sanitation [23, 25]. To date, limited research has assessed the impact of water interventions on mental health outcomes [26], and few studies have assessed the impact of sanitation interventions on mental health outcomes [14].

Despite the urgent need to improve sanitation and hygiene—including the target of universal basic access to sanitation as part of Sustainable Development Goal target 6.2—many large-scale sanitation interventions have shown poor uptake and sustainability [27], as well as mixed impacts on health [1]. Without sustained sanitation and hygiene behavior change, health gains are unlikely.

Community-led total sanitation (CLTS) has been heralded as a low-cost approach to improve community coverage of sanitation [28]. CLTS uses a demand-side approach—promoting the demand to execute improved sanitation behaviors, rather than supply-side

provision of infrastructure—that involves engaging the community, typically via an initial "triggering" event, to become open defecation free (ODF) through community-activities and local champions. Rigorous evaluations of CLTS, like those of other sanitation interventions, have yielded mixed health effects [29–31]. To date, there is mixed evidence on the potential of CLTS to achieve and sustain changes to WASH coverage and access [32–35]. Engaging local champions in CLTS delivery may yield beneficial results. Program delivery through Health Extension Workers (HEW) and the engagement of teachers both led to substantial improvements in sanitation coverage and use [34], although less than when delivered by trained natural leaders in Ghana [32]; yet these gains were not well sustained [36].

There are several documented limitations of community-led total sanitation and hygiene (CLTSH), a variation of CLTS that incorporates hygiene-related interventions. HEWs charged with implementing CLTSH have many responsibilities, limited incentives and motivations, few tools, and little capacity to continually reinforce messages [37]. The use of negative affective motivators employed by CLTS(H) may not be culturally appropriate or the most effective drivers of sanitation and hygiene behavior change [38], and may erode mental well-being. Together, the focus on negative affective motivators, poor facilitation of initial triggering, and a lack of follow-up, has left many communities with negative impressions of CLTSH initiatives [39].

In Ethiopia, CLTSH, which has been implemented widely through the Ethiopian Health Extension Programme (HEP), relies chiefly on negative affective motives (e.g., shame, disgust) to drive open defecation cessation. However, like prior evaluations of CLTS, evidence suggests that CLTSH is largely ineffective, with one out of six Ethiopian households continuing to practice open defecation after their respective villages were certified as ODF [40].

We designed a study to generate evidence to address knowledge gaps related to demand-side sanitation and hygiene programming and examine less studied, yet critical, inter-personal factors related to sustained behavioral adoption and downstream health impacts [41]. Specifically, we conducted a cluster-randomized trial (CRT) to test whether an intervention delivered at scale within the existing Ethiopian HEP would lead to sustained WASH behavior change and improved mental health. Leveraging feedback received from community members and key stakeholders, we designed a theoretically-informed [42–44] and evidence-based demand-side sanitation and hygiene intervention called *Andilaye*—Amharic for "togetherness/integration." The intervention takes a positive, encouragement approach to behavior change by promoting incremental improvements in behavior and incorporating behavioral maintenance strategies to foster sustained behavior change.

## Methods

The study's primary aim was to determine whether a demand-side sanitation and hygiene intervention (*Andilaye*) impacted WASH behavior change and mental health, specifically general well-being and symptoms of anxiety, depression, and non-specific emotional distress. A protocol detailing the methods, intervention, and baseline results are published elsewhere [41].

### Ethics and trial registration

Ethical approval for the *Andilaye* Trial was provided by Emory University (IRB00076141), the London School of Hygiene & Tropical Medicine (9595), and locally by the Amhara Regional Health Bureau (HRTT0135909). The trial was registered with clinicaltrials.gov (NCT03075436) on March 9, 2017. Approval was obtained at district and sub-district government offices. Informed consent was obtained orally at each household due to low literacy rates

of the population and concerns about historically coercive practices which including obtaining signatures. Oral consent was approved by all ethics boards.

## Study design

This parallel CRT was conducted in West Gojjam and South Gondar Zones of the Amhara National Regional State, a region of Ethiopia in which WASH conditions are inadequate [45], slippage in sanitation coverage and improved sanitation behaviors has been documented, and several NTDs (e.g., soil-transmitted helminths, trachoma) are hyperendemic [46]. Three districts (*woredas*)—Bahir Dar Zuria *Woreda* in West Gojjam Zone and Fogera and Farta *Woredas* in South Gondar Zone—were targeted for this study and represent a range of the topographical conditions present in Amhara, and Ethiopia in general. We targeted specific behaviors, including sanitation (constructing, maintaining, and using a latrine), personal hygiene (handwashing at key times and face washing), and household environmental sanitation (keeping animals separate from living quarters and keeping the compound free of feces). We sought to investigate whether any changes in WASH behaviors targeted by the *Andilaye* intervention were sustained, and we tracked intervention fidelity through a process evaluation.

We employed a structured sampling strategy to randomly select 50 eligible clusters within our sampling frame. Fig 1 provides further details in the CONSORT flow diagram. Clusters were defined as rural or peri-urban sub-districts (*kebeles)*—the smallest government administrative unit in Ethiopia—that were accessible throughout the course of the year. Of the 50 *kebeles* enrolled into the study, 22 were selected from Farta, 12 from Fogera, and 16 from Bahir Dar Zuria. The secondary sampling unit for this study was the household; specifically, any household residing in a targeted, sentinel village (*gott*) within a randomly selected study *kebele*. We utilized a 'fried egg' [47] approach to purposively select one to two *gotts* that were either situated in or near the center of the *kebele* (if there were centric *gotts*) or were not adjacent to any other study *kebele* (in the event there are no centric *gotts*). This approach minimized spillover of intervention effects and other externalities associated with the research between intervention and control clusters, especially those adjacent to each other. The number of targeted *gotts* depended only on the number of eligible households identified in *gott* census books.

Following baseline data collection, a stratified random design at the *woreda*-level was used to assign an equal number of study *kebeles* to either the *Andilaye* intervention or the control group receiving no intervention using a computer-based random number generator. To secure balance across three key potential confounders (i.e., latrine coverage, washing station with soap coverage, and head of household education), we established *a priori* that the intervention and control mean values for these three variables, using baseline data, should be within two standard deviations of the overall mean [41]. The randomization process was repeated twice using replacement rerandomization [48] to achieve balance according to that *a priori* criterion.

The Health Extension Services Package, and its accompanying CLTSH module delivered via the HEP, were being scaled throughout Ethiopia [49] and reflected the existing government-supported demand-side sanitation and hygiene approach. No attempt was made to modify the roll out of this standard of care of WASH programming in any of our study *kebeles*. There were no meaningful differences in the number of previously CLTSH-triggered and ODF certified *kebeles*, between study arms (Fig 1). Baseline WASH and demographic statistics [41], along with the fact that 39 of 50 *kebele* study clusters randomly selected for inclusion in the *Andilaye* Trial had been triggered with CLTSH, and certified ODF, provided strong evidence that behavioral slippage was, indeed, an issue that needed to be addressed in Amhara and perhaps elsewhere in Ethiopia.

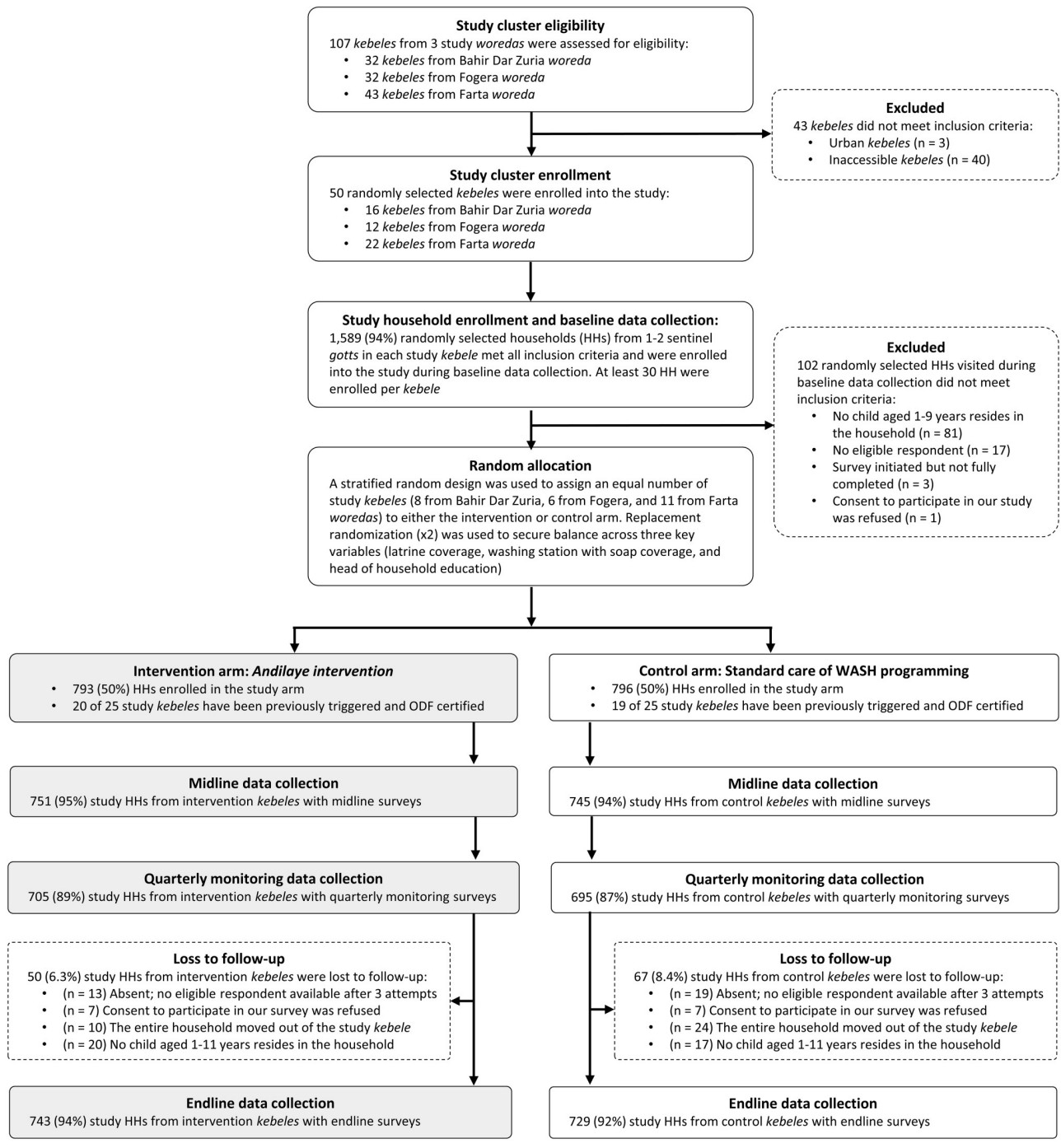

**Fig 1. CONSORT flow diagram.**

## "Andilaye" intervention

The *Andilaye* intervention motto was "*Together we can be a strong, caring, healthy community*". Intervention activities offered aspirational messages that emphasized the need for collective action to make positive change in the community and used verbal persuasion to enhance

collective efficacy perceptions [50]. The *Andilaye* intervention focused on three WASH-related behavioral themes, informed by formative research: (1) sanitation, (2) personal hygiene, and (3) household environmental sanitation. Within these themes were 11 constituent practices targeted by the intervention (Table 1); these practices were identified through formative research as ones that could be targeted using demand-side approaches, and were seen as achievable, per stakeholder feedback [41].

Intervention activities and behavior change tools were informed by our formative research and specifically designed to incorporate techniques that addressed behavioral factors such as action knowledge, perceived personal and household barriers to behavioral adoption, identification and planning, and behavioral control perceptions amongst others [41]. Activities occurred at four levels—district, community, group, and household (S1 Fig and S1 Table). Key activities included community mobilization and commitment events, community conversations with influential community members through facilitated group dialog, and household counseling visits with caregivers—all of which were guided by behavior change tools (e.g., community commitment banner, community conversations flipbook, and household counseling flipbook and goal cards) with illustrations produced by an artist based in Ethiopia.

The *Andilaye* intervention was delivered through Ethiopia's HEP, via trained government-salaried *Woreda* Health Office officials, HEWs, and volunteer Women's Development Army Leaders (WDALs) (Fig 2; see S2 Table for an alignment of relevant roles and responsibilities of the HEP and *Andilaye* Trial). Implementation of the *Andilaye* intervention was overseen by an Ethiopia-based study team. The *Andilaye* intervention commenced with district-level capacity building activities, such as action planning to orient key government stakeholders and training of trainers who would facilitate intervention activities in *kebeles* allocated to receive the *Andilaye* intervention. Further, district-level refresher trainings and adaptive management activities were conducted to reinforce previously acquired knowledge and skills, address trainer/facilitator turnover, and review successes and address challenges faced in implementing group and household level activities. Community-level activities included the 'Whole System in the Room' [51], community mobilization and commitment events, and cross-fertilization visits. These activities intended to engage community stakeholders in action planning, create an enabling environment in which change may occur, and address inter-personal factors related to public commitment, social norms, and social support related to improved practices, among others. WDALs from each intervention *kebele* were trained on how to conduct *Andilaye* household counseling visits with caregivers from each household in her catchment area. WDALs are unpaid community health workers as part of the government-organized Women's Development Army (WDA) strategy, which uses networks of neighboring women to increase the efficiency of HEWs in reaching every household, with one WDAL for every 30 households. During household-level counselling visits, trained WDALs provided personalized counselling to caregivers to equip them with the knowledge, skills, and motivation necessary to adopt and maintain improved WASH practices. Structured community conversations, implemented by trained community facilitators, provided further opportunity for group-level counselling and support.

## Outcomes of interest

Survey instruments administered for our impact evaluation collected data on key outcomes through self-reports from respondents and other household members. Primary outcomes of interest included mental health and three targeted WASH behavioral themes (1) sanitation, (2) personal hygiene, and (3) household environmental sanitation behaviors, consisting of 11 constituent practices (Table 1). To measures WASH outcomes, we pulled from standard WASH

**Table 1. Key outcome indicators for WASH behavioral themes and constituent practices of interest of the *Andilaye* intervention at endline.**

| Indicators | Intervention | | Control | | | |
|---|---|---|---|---|---|---|
| Sanitation (S) | Total N | % | Total N | % | PR (95% CI) [a] | PD (95% CI) [b] |
| *S1*: Construct a long-lasting latrine that is comfortable and hygienic | | | | | | |
| • Households with access to at least one household latrine | 743 | 61.2 | 729 | 62.0 | 0.99 (0.82, 1.21) | 0.00 (-0.13, 0.12) |
| • Households with access to an improved household latrine [c] | 741 | 34.6 | 726 | 30.6 | 1.13 (0.81, 1.59) | 0.41 (-0.07, 0.15) |
| • Households with access to a fully constructed household latrine | 742 | 33.0 | 729 | 28.7 | 1.15 (0.86, 1.54) | 0.04 (-0.46, 0.13) |
| *S2*: Repair your latrine whenever it is damaged | | | | | | |
| • Facility observed to require obvious repair | 455 | 70.1 | 451 | 80.5 | **0.88 (0.78, 0.99)** | **-0.10 (-0.19, -0.01)** |
| *S3*: Upgrade your latrine so it becomes more long-lasting, comfortable, and hygienic | | | | | | |
| • Household has added or improved anything on the latrine since its original construction | 453 | 17.2 | 446 | 15.7 | 1.08 (0.71, 1.65) | 0.01 (-0.06, 0.08) |
| • Households with latrine with smooth and cleanable slab/floor | 743 | 16.3 | 728 | 13.3 | 1.19 (0.70, 2.03) | 0.03 (-0.05, 0.11) |
| • Presence of drop hole cover in the latrine | 455 | 18.2 | 451 | 10.0 | **1.77 (1.19, 2.63)** | **0.08 (0.02, 0.14)** |
| *S4*: Close your pit when it becomes full and reconstruct a new latrine | | | | | | |
| • Is the pit that is in use full or close to being full | 454 | 11.7 | 451 | 12.6 | 0.92 (0.57, 1.49) | -0.01 (-0.68, 0.05) |
| *S5*: All household members use a latrine every time they defecate | | | | | | |
| • Respondent always exclusively used a latrine for defecation during last 7 days | 743 | 53.2 | 729 | 54.1 | 0.99 (0.79, 1.24) | 0.00 (-0.12, 0.12) |
| • Head of household always exclusively used a latrine for defecation during last 7 days | 529 | 36.5 | 473 | 33.0 | 1.07 (0.79, 1.47) | 0.03 (-0.09, 0.15) |
| • Ages 4–17 always exclusively used a latrine for defecation during last 7 days | 1447 | 42.6 | 1385 | 35.0 | 1.15 (0.89, 1.50) | 0.06 (-0.05, 0.16) |
| *S6*: Immediately dispose of children's feces into the latrine | | | | | | |
| • Child feces were safely disposed of during the last 2 days | 401 | 36.7 | 376 | 41.2 | 0.96 (0.69, 1.32) | -0.02 (-0.15, 0.11) |
| **Personal hygiene (PH)** | | | | | | |
| *PH1*: All household members wash their hands with water and soap or soap substitute AFTER handling animal and human feces, even children's feces | | | | | | |
| • Household hand or facewashing station(s) | 743 | 98.3 | 729 | 97.7 | 1.01 (0.99, 1.02) | 0.01 (-0.01, 0.02) |
| • The last time the respondent defecated, s/he cleaned hands with water and soap, substitute | 738 | 51.9 | 725 | 46.1 | 1.13 (0.94, 1.35) | 0.06 (-0.03, 0.15) |
| • The last time the index child defecated, s/he cleaned hands with water and soap, substitute | 713 | 43.9 | 697 | 39.6 | 1.12 (0.92, 1.35) | 0.05 (-0.04, 0.13) |
| *PH2*: All household members wash their hands with water and soap or soap substitute BEFORE handling food | | | | | | |
| • The last time the respondent prepared food, s/he cleaned hands with water and soap, substitute before beginning food preparations | 700 | 53.6 | 703 | 48.5 | 1.11 (0.95, 1.29) | 0.05 (-0.03, 0.13) |
| *PH3*: All household members wash their faces with water whenever they are dirty and use soap when it is available | | | | | | |
| • Ocular discharge present among children aged 1–9 years | 822 | 26.9 | 874 | 30.4 | 0.88 (0.68, 1.15) | -0.04 (-0.11, 0.04) |
| • Wet nasal discharge present among children aged 1–9 years | 822 | 37.0 | 874 | 39.4 | 0.94 (0.78, 1.13) | -0.02 (-0.09, 0.05) |
| • Dry nasal discharge present among children aged 1–9 years | 822 | 42.7 | 874 | 45.2 | 0.97 (0.81, 1.16) | -0.02 (-0.10, 0.06) |

*(Continued)*

**Table 1.** (Continued)

| Indicators | Intervention | | Control | | | |
|---|---|---|---|---|---|---|
| **Sanitation (S)** | **Total N** | **%** | **Total N** | **%** | **PR (95% CI) a** | **PD (95% CI) b** |
| • Dirt/dust/other debris present among children aged 1–9 years | 822 | 50.5 | 874 | 49.5 | 1.03 (0.89, 1.20) | 0.02 (-0.06, 0.09) |
| **Household Environmental Sanitation (HES)** | | | | | | |
| **HES1**: *Keep all animals separated from the house* | | | | | | |
| • Observed animal feces present in the compound | 743 | 82.2 | 729 | 82.4 | 1.01 (0.92, 1.11) | 0.01 (-0.07, 0.08) |
| **HES2**: *Keep the household compound clean by disposing of all animal feces and other waste on a DAILY basis* | | | | | | |
| • Animal feces/waste not left out in open in compound | 743 | 56.4 | 729 | 51.2 | 1.10 (0.95, 1.28) | 0.05 (-0.03, 0.13) |
| • Solid waste was not observed to have been left out in the open | 743 | 34.6 | 729 | 27.6 | 1.26 (0.93, 1.69) | 0.07 (-0.02, 0.17) |

Notes.

[a] We used log-linear binomial regression models to compare the prevalence of the outcomes between the intervention and control arms. Models accounted the stratified design by including woreda indicator variables [58], and accounted for clustering within kebeles by using generalized estimating equations with robust standard errors.

[b] Prevalence differences (PD) were calculated using post-estimation commands to estimate the average marginal effects.

[c] "Improved" was defined based on the WHO/UNICEF Joint Monitoring Programme (JMP) for Water Supply and Sanitation definition.

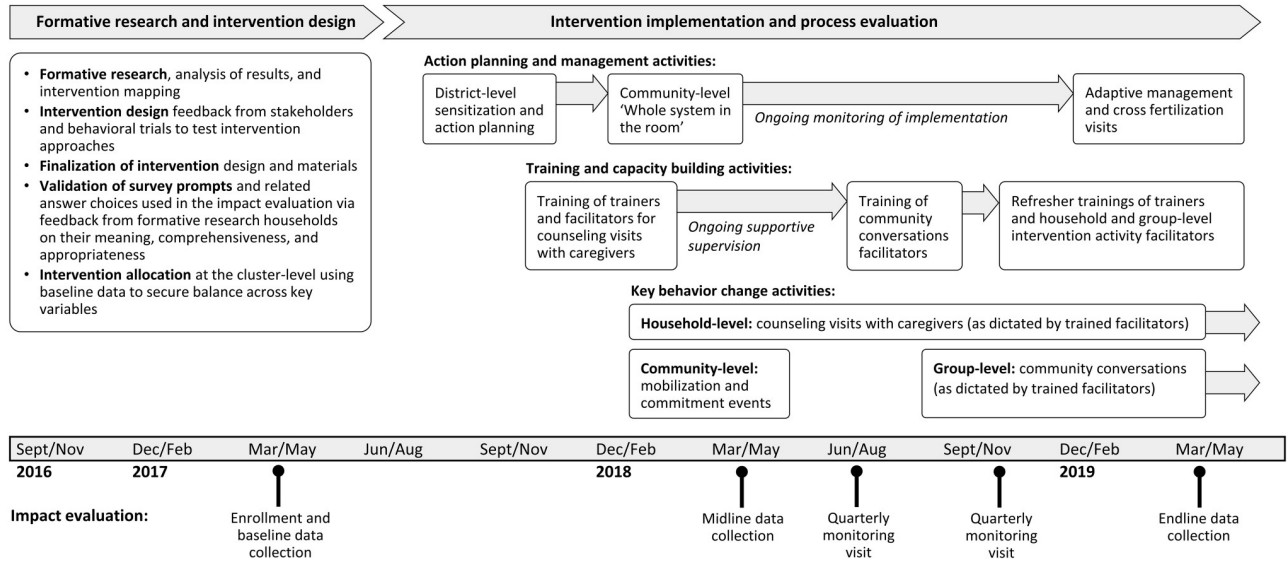

**Fig 2. The *Andilaye* Trial consists of three major phases: (1) formative research and intervention design, (2) intervention implementation and process evaluation, and (3) impact evaluation.** *Kebele* and household enrollment took place during baseline data collection (March to April 2017). Implementation of *Andilaye* intervention activities began in September 2017 and continued through midline data collection (March to April 2018), quarterly monitoring (June to July and November to December 2018), and endline evaluation (March to May 2019). See S1 Table for specific dates of the delivery of intervention activities. Midline data reflected at least 2 months since the start of household-level behavior change activities and 3 weeks since the completion of a catalyzing community-level mobilization and commitment event. Our endline data reflected the implementation of 14–15 months of household-level behavior change activities and 6–7 months of group-level behavior change activities (as dictated by trained activity facilitators) and 13–14 months since the community mobilization and commitment events.

indicators and leveraged formative research data to contextually adapt survey prompts and answer choices (shown in S4 Table of Delea et al., 2019). Sustainability of WASH-related behaviors was measured through the proportion of individuals and households consistently practicing target behaviors at midline and endline.

For mental health, we assessed subjective well-being using the validated WHO's Well-Being Index (WHO-5) [52] and symptoms of anxiety, depression, and non-specific emotional distress using the Hopkins Symptom Checklist (HSCL) [53]. WHO-5 asks the respondent to indicate how frequently they relate to each of five statements in the previous two weeks using a five-point Likert scale. Higher scores are better (range: 0–25) with scores below 13 indicating poor well-being. The HSCL is a non-diagnostic tool that includes 25 items to assess symptoms of anxiety (items 1–10), depression (items 11–25) and overall emotional distress (all 25 items). We omitted two items from the depression set: an item on sexual desire, which was deemed inappropriate for unmarried women, and an item on suicide ideation, because we were unable to provide clinical recourse if needed. Participants indicated how much symptoms bothered them in the previous week ('not at all' [1] to 'Extremely' [4]). The final score for each state is a mean of responses for each of the relevant items (range from 1 to 4). Scores of 1.75 or higher indicate that the condition could be present while lower scores are an indication of lower anxiety, depression, or distress.

Secondary outcomes included 7-day and 2-day diarrhea period prevalence, measured through caregiver report of the index child (i.e., youngest child in the household aged one to nine years at baseline), sanitation insecurity, and water insecurity. For sanitation insecurity, we asked respondents to indicate how often (never, sometimes, often, always) they felt one of seven different forms of sanitation insecurity (i.e., 7 factors). Scores were means of all items in the factor. A higher score represents higher sanitation insecurity. The factors were predesignated, and based on a validation that was done in another study [23]. Water insecurity was measured through the Household Water Insecurity Experiences (HWISE) scale [54]. HWISE includes 12 items with four response categories (never, rarely, sometimes, often/always). The score is the sum of responses, ranging from 0–36. A higher score indicates greater household water insecurity, and we considered households to be water insecure with a score of 12 or more, as described elsewhere [54]. Measures of collective efficacy [50] and intermediate behavioral antecedents were also collected, but are not reported here.

## Process evaluation

We conducted a process evaluation alongside our impact evaluation to describe and analyze key aspects of the *Andilaye*'s implementation and provide insights and understanding of program impacts. We defined fidelity as the degree to which the intervention or program was delivered as intended [55]. Quantitative process data on dose delivered, participation, and dose received were collected through the direct observation of all district and community-level activities—these activities being facilitated or co-facilitated by the Ethiopia-based study team. Questions were incorporated into our survey instruments administered during quarterly monitoring and endline data collection to capture exposure to key *Andilaye* behavior change activities by respondents from study-enrolled households in intervention *kebeles*. This included self-reported awareness of and attendance at the community mobilization and commitment event and community conversations, and the number of household counseling visits received from WDALs. Per protocol, all community members were targeted to attend the community mobilization and commitment event; routine (1–2 per month) community conversations primarily focused on influential community members (e.g., male heads of households, religious leaders, mother-in-laws) targeted in the 'Whole System in the Room'; and caregivers were to receive

monthly household counseling visits (each visit lasting approximately 30 minutes) following the typical structure for the WDAL and in accordance the HEP.

## Sample size and power

A detailed description of sample size considerations is published in the study protocol [41]. Briefly, we powered this study on mental well-being outcomes, as measure by the HSCL [53], utilizing data from Ethiopia and East Africa suggesting that approximately 20–35% of rural women experience elevated symptoms of anxiety and depression [56, 57]. Our sample size determination indicated we should recruit and enroll 25 households from each of our 50 study *kebeles*, with 25 *kebeles* per study arm, targeting one index child per household. We increased our final sample size to accommodate for 20% of households being lost to follow-up. Our target sample, therefore, included 30 households in each *kebele*, or 1,500 households in total (i.e., 750 per study arm).

## Data collection

Data were collected via structured household interviews and observations by trained enumerators during rounds of data collection. Surveys were collected using mobile phones equipped with the freely available Open Data Kit (http://opendatakit.org/). Households with at least one child aged one to nine years were randomly selected from the *gott* census book residing in the target *gott*(s) of the 50 study *kebeles* at baseline, and were followed for each round of data collection. Fig 2 and S1 Table provide a summary of the timeline of intervention implementation in relation to points of data collection for our impact evaluation. At the time of endline data collection, household-level activities had taken place for the past 14–15 months, group-level activities for the past 6–7 months, and community-level mobilization and commitment events were completed 13–14 months prior. Elements of the intervention were still ongoing during the time of our endline data collection, and we did not collect further data after the endline visit. Given the nature of the intervention, neither participants nor field teams were blinded to intervention status.

## Analytical methods

We followed a pre-analysis plan developed following baseline data collection [41]. The primary analysis method was an "intention-to-treat" analysis, which compares the intervention arm to the control arm without regard to intervention fidelity or compliance. The majority of our primary and secondary outcomes were binary variables, and for these we used log-linear binomial regression models and report the prevalence ratio (PR). For these binary outcomes we also present prevalence differences (PD), which were calculated using the post-estimation margins command to estimate the average marginal effects. For continuous outcomes, such as WHO-5 and HSCL scores, we used linear regression models. All models included an intervention variable as a fixed effect, accounted for the stratified design through the inclusion of the *woreda* indicator variable [58], and incorporated generalized estimating equations with robust standard errors to account for the clustering of observations within *kebeles*. For each of our primary outcomes of interest, we assessed if there was interaction across various sub-groups, including exposure to previous CLTSH triggering and sex of the index child. We also assessed if water insecurity modified the effectiveness on hygiene behaviors. For all of these analyses, we included interaction terms to test if effect modification was present (i.e., the interaction term had a p-value <0.05).

To assess whether any improvements in WASH behaviors were sustained between follow-up periods, we compared the prevalence of key targeted sanitation, hygiene, household

environmental sanitation indicators between the demand-side intervention arm and the control arm group using the baseline, midline, and endline data (Fig 2).

## Results

### Survey results

Our baseline assessment showed balance in terms of our primary outcomes of interest and demographic variables [41]. Our endline results reflect complete data from 1,472 (93%) of 1,589 households enrolled in the study at baseline, and exceeded our sample size requirement of 1,250 households. Of the 793 enrolled intervention households and 796 enrolled control households, retention was similar in both arms, at 94% and 92%, respectively (Fig 1). A large majority (90%) of the respondents were female, who were typically the primary caretakers. Of these 1,472 respondents, 85% were the mother of the index child.

### Process evaluation

Reports from *Woreda* Health Offices collected at endline indicated that none of the 50 study *kebeles* (intervention or control) received additional CLTSH triggering or re-triggering during the course of the *Andilaye* Trial. For the *Andilaye* intervention, the fidelity of action planning workshops and trainings at the district and community levels were high (See S3 Table for a summary of process data on dose delivered, participation, and dose received for all intervention activities). All three study *woredas* and their intervention *kebeles* (n = 25) had action planning and management (S3a Table) and training and capacity building (S3b Table) intervention activities completed as planned, at the district and community levels, respectively. Participation was high among targeted government and community stakeholders during catalyzing and maintenance action planning and management activities (S3a Table). Nearly all of our targeted *Woreda* Health Office officials, HEWs, and WDALs were trained on *Andilaye* counseling visits with caregivers and all intervention *kebeles* had a pair of facilitators trained on *Andilaye* community conversations—including rounds of review meetings and refresher trainings (S3b Table). All intervention *kebeles* had a community mobilization and commitment event completed as planned, with an estimated average of around 300 adult community members in attendance per event (S3c Table).

Our household process evaluation survey results reflect complete data from 703 (89%) and 707 (89%) of 793 study-enrolled households in intervention *kebeles* from quarterly monitoring and endline data collection, respectively. Reported frequency of household-level counseling visits reflected a total of 665 study-enrolled households as these questions were not relevant for 42 households that were residents of WDALs who were trained to conduct *Andilaye* counselling visits.

Household respondent-reported exposure of key behavior change activities varied, but was generally suboptimal (Fig 3). Overall, only 18% of respondents reported attending the community mobilization and commitment event, and 22% reported being aware of the activity in the months following the event (i.e., during quarterly monitoring). At endline, 28% of respondents reported attending at least one community conversation, and 46% reported being aware of the activity. WDALs and their supervisors (i.e., HEWs) were trained to facilitate monthly counseling visits with households in their catchment area. However, at endline, only 59% of respondents reported receiving a counseling visit and 43% reported receiving at least one follow-up visit during the 14-15-month implementation period. No intervention *kebele* had WDALs conducting counseling visits monthly. The average number of visits was 2–3 among respondents reporting at least one counseling visit (n = 391) (S3c Table). For households receiving a visit, 72% of respondents reported that they set household goals or incremental improvements,

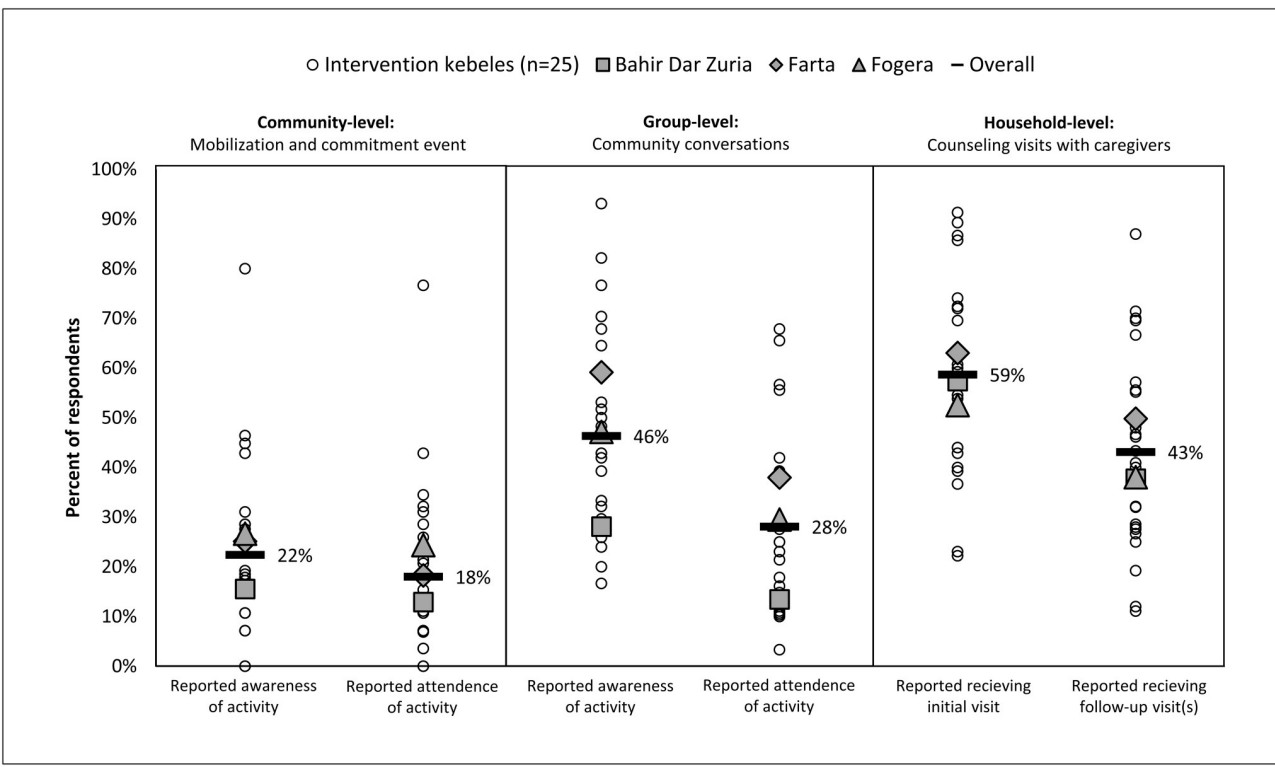

**Fig 3. Respondent-reported exposure of key behavior change activities of the *Andilaye* intervention.** Respondents from study-enrolled households in intervention *kebeles* (n = 793) were surveyed on their awareness and attendance in the community-level mobilization and commitment event during quarterly monitoring visits and awareness and attendance of group-level community conversations and frequency household-level counseling visits received by endline visits. A total of 703 (89%) and 707 (89%) surveys with responses to process evaluation prompts were completed from quarterly monitoring and endline, respectively. Reported frequency of counseling visits reflected a total of 665 survey responses as these questions were not relevant for study-enrolled households that were residents of caregivers who were trained as Women's Development Army Leaders (WDALs) responsible for conducting the *Andilaye* counselling visits (n = 42).

and two-thirds reported that they identified barriers and their WDAL provided counseling on how to plan for, cope with, and overcome barriers in accordance to the *Andilaye* behavior change tools (i.e., household counseling flipbook and goal cards) (S3c Table).

## Impacts on sanitation, personal hygiene, and household environmental sanitation

The intervention did not increase latrine access. At endline, 62% of both intervention and control households had at least one latrine (prevalence ratio [PR] 0.99; 95% CI: 0.82, 1.21) (Table 1). There was no difference in the prevalence of improved latrines (PR 1.13; 95% CI: 0.81, 1.59) or in fully constructed latrines (PR 1.15; 95% CI: 0.86, 1.54). Although there were improvements in many latrine characteristics in the intervention arm compared to the control arm (e.g., presence of water available or cleansing agent near or inside the latrine for hand-washing, and water available for flushing or self-cleansing), the conditions (e.g., presence of feces on floor) of latrines in the intervention arm were often poor (S4 Table).

The intervention did not impact defecation practices. Overall, 40% of respondents reported practicing open defecation during the previous two days; only 46% of respondents had defecated in any latrine during the previous two days (S4 Table). All measures of latrine utilization and non-utilization were similar across intervention and control arms. This includes

indicators of respondent open defecation (PR: 1.05; 95% CI: 0.76, 1.45), safe disposal of child feces (PR: 0.96; 95% CI: 0.69, 1.32), and number of people from another household who used a latrine during last seven days (difference: -0.40; 95% CI: -0.85, 0.05).

The intervention did not impact personal hygiene behaviors. The prevalence of washing stations with water (PR: 0.96; 95% CI: 0.72, 1.26) was similar between the intervention and control arms. Presence of hand or face washing stations were observed in 98% of households (Table 1), although water and soap were observed in only 20% of handwashing stations and 2% of facewashing stations (S4 Table). The prevalence of stations with soap was higher in the intervention arm, although only 3% of households in this arm had a washing station with soap present. Among all children aged one to nine years at endline, observations of facial cleanliness indicated 29% had ocular discharge, 38% had wet nasal discharge, 44% had dry nasal discharge, and 50% had dust, dirt, or debris on their faces (Table 1). There were no meaningful differences between the study arms for any of these facial cleanliness measures.

We found no evidence that the *Andilaye* intervention impacted household environmental sanitation. Across both arms, the majority of respondents and heads of household had animal herding responsibilities (88% overall), and animal feces were present in the compound in 82% household compounds (S4 Table). A similar proportion of households in intervention and control kept animals separate from the house (PR = 1.01, 95% CI 0.91, 1.11). About half of households did not leave animal feces/waste in the open (Table 1); this was similar between the intervention and control arms (PR = 1.10; 95% CI: 0.95, 1.28).

**Sustained changes of key indicators.** There was little difference in the sustainability of key targeted indicators—assessed by changes between midline and endline—on sanitation access and practices, personal hygiene access and practices, and household environmental sanitation over the course of follow up (Fig 4). At midline, most variables continued to show little difference between the intervention and control arms, although the prevalence of drop hole covers in latrines and the prevalence of appropriate hygiene behaviors were more common in the intervention arm. At the endline visit, the prevalence of drop hole cover was largely sustained in the intervention arm, while the prevalence of drop hole covers decreased in the control arm (PR = 1.77' 95% CI: 1.19, 2.63). All other variables at endline had similar prevalence levels when comparing the two arms. While the prevalence of hand hygiene behaviors was maintained at levels similar to the midline visit, increases in hand hygiene behaviors in the control arm narrowed the difference between the intervention and control arms at endline. Over the two follow-up surveys, there was an increase in the prevalence of household hand or facewashing stations that appeared among study arms.

## Impacts on mental health

There was no difference between study arms in the scores for anxiety, depression, emotional distress or general well-being (Table 2). There was also no difference between the intervention and control arms in the prevalence of each mental health condition: anxiety (PR = 0.90; 95% CI: 0.72, 1.11), depression (PR: 0.83; 95% CI: 0.64, 1.07), emotional distress (PR: 0.86; 95% CI: 0.67, 1.09) and poor well-being (PR: 0.90; 95% CI: 0.74, 1.10) (Table 2). All measures of mental health trended in the protective direction for both mean scores and prevalence (2–3% reduction), but were not statistically different between study arms.

## Secondary health outcomes

**Reported diarrhea.** Diarrhea prevalence during the last seven days among index children was similar in the intervention (7%) and control (6%) arms (PR: 1.20; 95% CI: 0.74, 1.93;

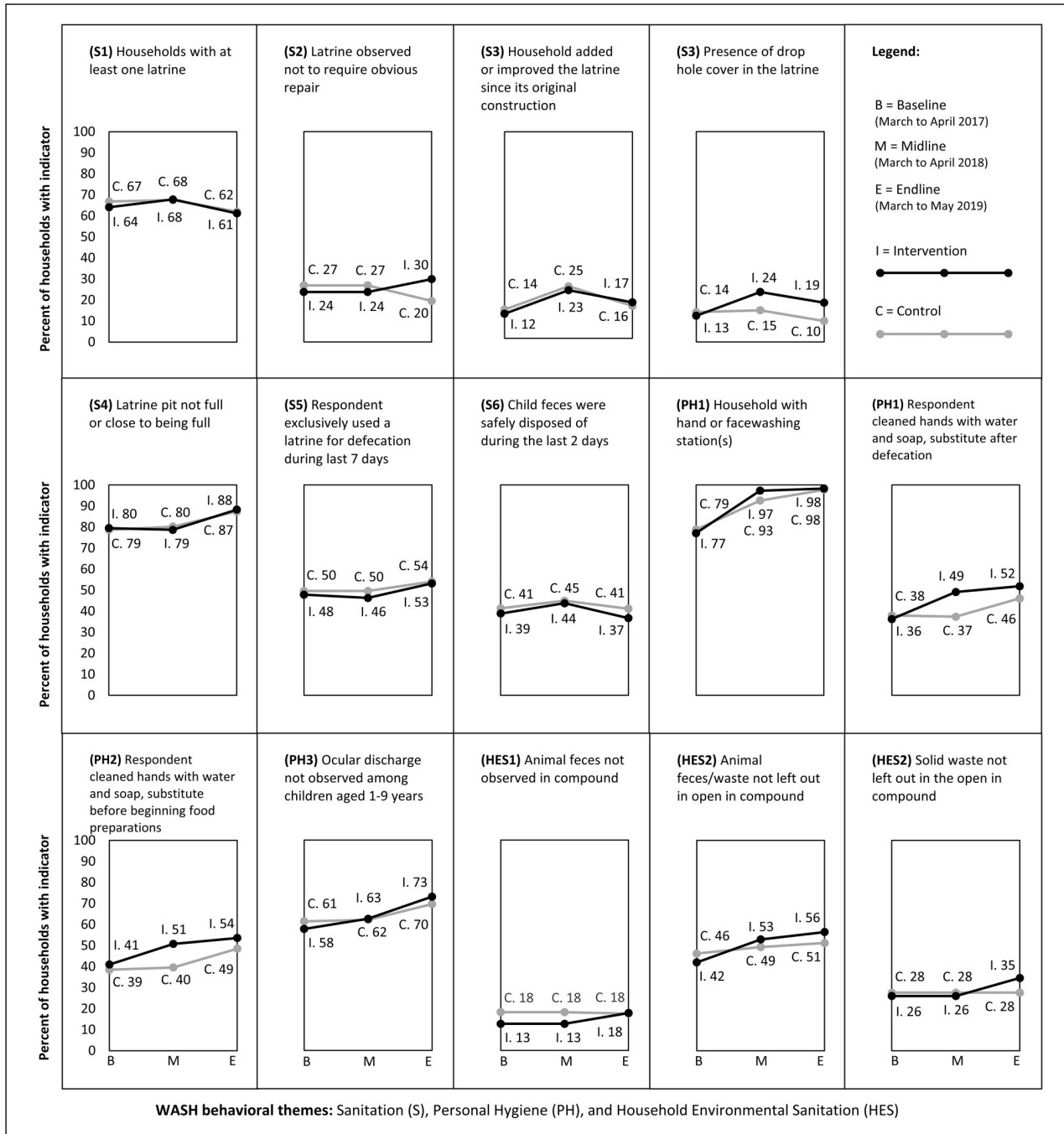

**Fig 4. The prevalence of key sanitation, personal hygiene, and household environmental sanitation indicators over time.**

Table 3). Among index children, there were also similarities comparing study arms in diarrhea prevalence when assessing episodes over the last two days (PR: 1.25; 95% CI: 0.71, 2.22).

**Water and sanitation insecurity.** The intervention did not statistically reduce water insecurity prevalence between intervention (5.7%) and control (8.8%) arms (PR: 0.50 (95% CI: .21, 1.2); Table 3). At endline, sanitation insecurity scores related to social support were statistically lower (i.e., better) in the intervention arm than in the control arm (score difference: -0.10, 95%

**Table 2. Mental well-being outcomes at endline.**

| Indicator | Cronbach Alpha | Intervention | | Control | | | |
|---|---|---|---|---|---|---|---|
| Scores | | N | Mean (SD) | N | Mean (SD) | - | difference (95% CI)[ae] |
| Anxiety score [b] | 0.89 | 742 | 1.46 (0.61) | 729 | 1.52 (0.64) | - | −0.06 (−0.14, 0.02) |
| Depression score [b] | 0.87 | 742 | 1.35 (0.48) | 728 | 1.39 (0.52) | - | −0.04 (−0.08, 0.01) |
| Emotional distress score [b] | 0.93 | 741 | 1.29 (0.46) | 728 | 1.33 (0.49) | - | −0.04 (−0.09, 0.01) |
| Well-being score [c] | 0.97 | 749 | 17.6 (6.8) | 728 | 17.0 (6.7) | - | 0.50 (−0.19, 1.28) |
| Prevalence | | N | % | N | % | PR (95% CI) [d] | PD (95% CI) [e] |
| High Anxiety [f] | - | 742 | 22.2 | 729 | 24.8 | 0.90 (0.72, 1.11) | −0.03 (−0.08, 0.03) |
| High Depression [f] | - | 742 | 14.0 | 728 | 16.9 | 0.83 (0.64, 1.07) | −0.03 (−0.07, 0.01) |
| High Emotional distress [f] | - | 741 | 14.0 | 728 | 16.4 | 0.86 (0.67, 1.09) | −0.02 (−0.06, 0.01) |
| Poor well-being [g] | - | 749 | 25.2 | 728 | 27.8 | 0.90 (0.74, 1.10) | −0.03 (−0.08, 0.02) |

Notes.

[a] We used linear regression models to estimate the difference in the outcomes comparing the intervention and control arms. Models accounted the stratified design by including woreda indicator variables [58], and accounted for clustering within kebeles by using generalized estimating equations with robust standard errors.

[b] We asked respondents to indicate how much the symptoms bothered them in the previous week with four potential response options (not at all (1) to extremely (4)). The first ten symptoms assess anxiety (i.e., 'suddenly scared for no reason', 'nervousness or shakiness inside'), the next 13 assess depression (i.e. 'feeling low in energy', 'feeling hopeless about the future'), and the 23 collectively assess non-specific emotional distress. For each outcome, the score is the sum of the responses divided by the number of items.

[c] We asked respondents about well-being, and responses ranged from '(0) At no time' to (5) All of the time'. Scores were summed, and range from 0–25; the higher the score, the better the well-being.

[d] We used similar log-linear binomial regression models to compare the prevalence of the outcomes between the intervention and control arms.

[e] Prevalence differences (PD) were calculated using post-estimation commands to estimate the average marginal effects.

[f] Each of the above scores was dichotomized, with scores greater than 1.75 indicating a positive status for any of the three outcomes.

[g] The above score was dichotomized with scores below 13 indicating poor well-being.

CI: -0.16, -0.43), indicating a reduced frequency of experiencing the circumstances in the social support domain (e.g., trouble finding support to watch dependents during urination, worry about dependents when going to defecate, had to leave dependents alone to urinate, etc.). Other sanitation insecurity measures were similar between arms.

## Interaction and effect modification

There was no interaction of the intervention by previous CLTSH triggering for any of the primary outcome variables of interest. We did not detect effect measure modification by sex for any of the four mental health outcomes. Similarly, we did not detect interaction by child's sex for any of these outcomes. We also did not detect interaction between the intervention and water insecurity on any of the primary handwashing or face washing variables.

## Discussion

The *Andilaye* intervention generally did not improve WASH conditions or outcomes. Without sustained changes to these WASH conditions and behaviors, changes in well-being were not likely, and indeed, were not detected. Improving sanitation and hygiene behaviors in rural communities remains a considerable challenge, especially in regions with poor water access and high levels of WASH-related NTD endemicity. Most studies designed to change sanitation and hygiene behavior are *efficacy* studies [27]—meant to assess changes under controlled conditions; ours was an *effectiveness* study, designed to measure changes in a real-world context. We believe that poor fidelity of intervention delivery played a considerable role in uptake of

**Table 3. Secondary health outcomes at endline.**

| Indicator | Cronbach Alpha | Intervention | | Control | | | |
|---|---|---|---|---|---|---|---|
| **Diarrhea** | | N | % | N | % | PR (95% CI) [a] | PD (95% CI) [b] |
| During the last 2 days, index child had three or more loose stools per day | - | 730 | 5.9 | 720 | 4.7 | 1.62 (0.71, 2.22) | 0.01 (−0.018, 0.042) |
| During the last 7 days, index child had three or more loose stools per day | - | 731 | 7.1 | 721 | 6.0 | 1.20 (0.74, 1.93) | 0.01 (−0.019, 0.043) |
| **Water and sanitation insecurity scores** | | N | mean (SE) | N | mean (SE) | - | difference (95% CI) [c] |
| Water-HWISE Scale [d] | 0.96 | 565 | 1.71 (0.37) | 388 | 2.71 (0.89) | - | −1.29 (−3.19, 0.61) |
| Sanitation-Potential harms[d] | 0.85 | 365 | 0.46 (0.026) | 327 | 0.50 (0.033) | - | −0.05 (−0.13, 0.03) |
| Sanitation-Social expectations resultant repercussions [d] | 0.79 | 366 | 0.28 (0.025) | 327 | 0.30 (0.022) | - | −0.03 (−0.09, 0.03) |
| Sanitation-Physical exertion or strain [d] | 0.57 | 366 | 0.42 (0.046) | 328 | 0.40 (0.043) | - | 0.01 (−0.11, 0.13) |
| Sanitation-Night concerns [d] | 0.56 | 366 | 0.32 (0.022) | 328 | 0.37 (0.027) | - | −0.05 (−0.12, 0.02) |
| Sanitation-Social support [d] | 0.88 | 366 | 0.10 (0.021) | 328 | 0.20 (0.023) | - | **−0.10 (−0.16, −0.43)** |
| Sanitation-Physical agility [d] | 0.56 | 366 | 0.14 (0.017) | 328 | 0.14 (0.020) | - | 0.00 (−0.05, 0.05) |
| Sanitation-Defecation place [d] | 0.81 | 366 | 0.35 (0.038) | 327 | 0.32 (0.028) | - | 0.02 (−0.06, 0.11) |
| **Water insecurity prevalence** | | N | % | N | % | PR (95% CI) [a] | PD (95% CI) [b] |
| Water insecure (HWISE score 12 or more) [d] | - | 565 | 5.7 | 388 | 8.8 | 0.50 (0.21, 1.23) | -0.05 (-0.12, 0.03) |

Notes.

[a] We used log-linear binomial regression models to compare the prevalence of the outcomes between the intervention and control arms. Models accounted the stratified design by including woreda indicator variables [58], and accounted for clustering within kebeles by using generalized estimating equations with robust standard errors.

[b] Prevalence differences (PD) were calculated using post-estimation commands to estimate the average marginal effects.

[c] We used similar linear regression models to estimate the difference in the outcomes comparing the intervention and control arms.

[d] We asked respondents to indicate how often they felt some form of sanitation insecurity (never, sometimes, often, always). These items were then summed with all other items in that factor and divided by the numbers of items to create a score. The factors were predesignated, and based on a validation that was done in another study [23]. A higher score represents higher sanitation insecurity. [d] We used similar linear regression models to estimate difference comparing the outcomes between the intervention and control arms. This used a 12-item scale with four response categories (never, rarely, sometimes, often/always), and a total summed score of those response categories ranging from 0–36. A higher score indicates greater household water insecurity. We considered water insecure as a score of 12 or more, as described elsewhere [54].

the intervention, pointing to challenges in delivering demand-side sanitation and hygiene interventions at scale and through existing community-based models.

We did not find statistical differences between study arms at endline and few promising trends in intervention communities for some of the targeted behaviors. Changes were much lower than with approaches found by Crocker et al. elsewhere in Ethiopia [33, 36]. Similarly, Apanga et al. found that implementation of the *Rural Sustainable Sanitation and Hygiene for All* (SSH4A) approach that employs a multidimensional intervention led to a large increase of 77 percentage points in sanitation coverage in Ethiopia and also coverage gains in many other countries under study [59], although slippage did occur after conclusion of program activities, whereas many other countries sustained their previous sanitation coverage gains [60].

Our *Andilaye* intervention did not statistically impact validated mental health measures. Few studies have measured the impact of a sanitation intervention on mental health outcomes

despite calls for broader investigations of sanitation-related health impacts [14, 61, 62]. In rural India, women's experiences of sanitation, as measured by a validated sanitation insecurity measure, were associated with well-being, anxiety, depression, and distress, even when women had access to a facility [25]. Similarly, in urban Mozambique, latrine location and neighborhood violence were important determinants of safety perceptions and corresponding psychosocial stress [63]. These findings highlight the need for interventions to consider the experience of sanitation beyond access to a facility alone and the intrinsic value of sanitation [64]. We assessed if changes to sanitation access and sanitation insecurity—changes that we anticipated would be generated by this intervention—would lead to improved mental health states, including improved well-being and reduction in symptoms associated with anxiety, depression, distress and general wellbeing. The intention-to-treat analysis did not detect changes to mental well-being scores or to sanitation insecurity scores, which was perhaps limited by our short evaluation period, and sanitation quality (Fig 2). However, we saw some, non-statistically different, preventive trends across all mental health measures which were primarily among women (90% of respondents). Our impact on sanitation social support is indicative of the underlying philosophy of the *Andilaye* intervention, which was designed contrary to the "shame" drivers of more traditional CLTSH [28, 65]. These findings are encouraging despite the low fidelity of the intervention delivery at household-level, suggesting improved fidelity may result in evidence of impact. We believe further studies are warranted to test the hypothesis that improved sanitation would impact mental well-being, as our intervention did not change sanitation behaviors, quality, or access.

The purpose of this study was to develop and test an intervention that could be scaled within the existing Ethiopian HEP. The intervention was designed to be incorporated into prevailing programs (e.g., HEP) to demonstrate potential for scale-up, and did not succeed in this regard. For example, despite high attendance at trainings and action planning workshops, and the provision of supportive supervision and on-the-job-training tools, many households did not receive *Andilaye* counseling visits (Fig 3). WDALs and HEWs reported that they did not receive supportive supervision from relevant government officials in accordance with their action plans (S3b Table). While supportive supervision considerations were acknowledged and incorporated into the design of the *Andilaye* intervention, these requirements did not go above and beyond what is expected of the HEP (S2 Table) [66]. These delivery challenges are consistent with those associated with CLTSH programming and HEP more broadly [37, 67]. Additionally, a majority of intervention *kebeles* had non-active WDALs at the start of implementation, as identified by our Ethiopian-based study team during initial recruitment of activity facilitators. At endline, only 66% of respondents from our process evaluation surveys were able to identify the WDAL responsible for conducting their *Andilaye* counseling visits (S3c Table). A cross-sectional study in four regions of Ethiopia found similar trends in varying levels of WDAL strategy implementation strength among 423 *kebeles* [68]. Importantly, findings from Damtew et al. suggest HEP outreach activities were higher in *kebeles* where active WDAL density was higher (i.e., fewer households per active WDAL). Although HEWs were paid health workers, WDALs were not. This has brought questions of ethics and sustainability as WDALs are increasingly asked to provide more and more services. Recent qualitative and quantitative studies suggest that unpaid WDALs are actually worse off than their peers and makes women, especially unmarried women, vulnerable to negative gossip and psychological distress [69, 70]. Although this point goes much deeper into the political economy, it is an important gap to bring up in the context of empowering women volunteers to enact positive change in their communities [71]. Further, when community health workers are paid to deliver the intervention, there is evidence of successful delivery [72]. Together, these findings

raise questions about the possibility of bringing new programs and approaches to the HEP without adequate support.

Evidence suggests that it is important to move away from information-based interventions to address the array of behavioral factors and determinants that operate at various levels of influence [73–78]. Few sanitation and hygiene interventions employ behavioral theory to locally adapt messages [36, 79]. Exceptions include the *SuperAmma* intervention, which was developed and implemented in India, and found substantial gains in handwashing with soap [80] and studies that examined the effectiveness of the risk, attitudes, norms, abilities and self-regulation (RANAS) behavioral model to intervention design and showed positive impacts on a variety of WASH behaviors including safe water consumption, solar water disinfection, handwashing, and cleaning of shared sanitation [81–87]. Our intervention aimed to focus on a variety of contextually appropriate behavioral factors rather than knowledge alone. Given the low fidelity of the intervention delivery, further capacity building of federal, regional, and local-level government officials as well as community-level change agents may be necessary for the successful implementation of approaches that move beyond dissemination of information and messages [88–90].

The WASH sector has traditionally relied on unpaid female labor (in this case HEWs, WDALs), which are gender exploitative approaches that reinforce women as the household duty bearers for WASH [91]. While our assessment of mental health highlights the potential impact on women beyond their roles as mothers and caregivers, our intervention strategy provided little exception to the longstanding program strategy of adding to the already burden-some roles as child caregiver. Interventions that fail to assess burdens on women, and mothers in particular, may impose harms or burdens that can exacerbate inequalities [92]. Gender-disaggregated data on the workload of women and girls in household responsibilities [93], as well as better sex-disaggregated data on program outcomes could support WASH strategies that lead to gender transformative, and ultimately more sustainable programming [61].

### Study strengths

Our intervention was theory-informed and included an extensive intervention design process during which we emphasized the solicitation and incorporation of feedback from key stakeholders at regional, zonal, *woreda*, and community-levels. It was designed to be delivered at scale within the Ethiopian HEP. We utilized a randomized study design, in which intervention and control communities were allocated to treatment arms randomly. While CRTs tend to emphasize internal validity, we made considerable effort to enhance external validity. Our study was spread over three *woredas* in two zones, yielding a heterogeneous mix of contexts and topographical conditions—which serve as a proxy for factors such as soil type, access to markets, and flooding risk—and support external validity of the findings. To improve interval validity, we used a 'fried egg' [47] approach—while allocation occurred at the *kebele* level, intervention activities and data collection occur in one to two sentinel *gotts* per *kebele*, purposively selected to minimize spillover. We targeted both rural and peri-urban communities and collected behavioral outcome data on a variety of household members (e.g., primary female caregiver of index child, head of household, all children aged 0–17 years).

### Study limitations

The study faced significant delays in gaining local ethical approval to start the project which led to truncated implementation and follow-up periods. Key government actors were less involved than planned, which may have led to sub-optimal fidelity. The integration of *Andilaye* intervention activities into non-Ethiopian HEP delivery structures (e.g., hired independent

community implementers) may yield further investigations into the effectiveness of the intervention on sustained behavior change and mental well-being. Several of our behavioral outcomes were reported (vs. observed), and these types of outcomes may be prone to reporting biases, indicated by differences in our reported and observed measures.

## Conclusions

We did not find that the *Andilaye* intervention yielded changes in behaviors and conditions related to sanitation, personal hygiene, or household environmental sanitation; nor did it impact mental health outcomes. Limited integration of *Andilaye* activities into the HEP likely explains the minimal impact observed and points to considerable challenges related to implementing demand-side interventions at scale in Ethiopia. There is a crucial need to identify and scale effective service delivery models in order to meet the ambitious Sustainable Development Goal targets for sanitation and hygiene [94]. Evidence from this trial may help address knowledge gaps related to scalable alternatives to CLTSH and inform sanitation and hygiene programming and policy in Ethiopia and beyond. A greater emphasis on implementation research in WASH delivery would support tools and approaches for developing, testing, and adapting scalable best-practice interventions [95].

## Supporting information

**S1 Fig. Diagram summarizing the *Andilaye* intervention.**
(PDF)

**S1 Table. Summary of the *Andilaye* intervention activities and dates of delivery.**
(PDF)

**S2 Table. Alignment of relevant roles and responsibilities of the Ethiopian Health Extension Programme (HEP) and *Andilaye* Trial.**
(PDF)

**S3 Table. Summary of process data for the *Andilaye* Trial.**
(PDF)

**S4 Table. *Andilaye* Trial indicators used to assess WASH behavior.**
(PDF)

## Acknowledgments

The authors would like to thank our *Andilaye* study participants, who generously gave their time to participate in our formative work, behavioral trials, and summative assessment surveys. We are grateful for the support we receive from numerous partners at the Democratic Republic of Ethiopia's Federal Ministry of Health; the Amhara Regional Health Bureau; South Gondar and West Gojjam Zonal Health Departments; and the Bahir Dar Zuria, Farta, and Fogera *Woreda* Health Offices. We would also like to acknowledge support provided by the Health Extension Workers, Women's Development Army Leaders, and Health Development Army members from our study *kebeles*. We thank the cadre of field supervisors and enumerators who captured these data (Yeworkwuha Abay, Mantegbosh Abebe, Selamawit Abebe, Tigist Abebe, Rosa Abesha, Mahider Adamu, Balemlaye Addisu, Adanech Admasu, Tirusew Alayu, Adisalem Arega, Yalemwork Asaye, Destaw Asnakew, Yalemwork Ayanew, Ayalnesh Belay, Asayech Bimrew, Tigist Bitew, Tiruzer Engidaw, Yeserash Gashaw, Natsenat Gebretsadkan, Woyneshet Genetu, Tewodaj Gizachew, Tibeltalech Mihiret, Yehizbalem Minale, Senait

Mulualem, Eleni Nebiyu, Elsabet Seyoum, Mulubirhan Shitu, Sewunet Tadesse, Beza Tesfaye, Rahel Tsegaye, Sintayehu Wasihun, and Maritu Yibrie).

## Author Contributions

**Conceptualization:** Matthew C. Freeman, Maryann G. Delea.

**Formal analysis:** Matthew C. Freeman, Jedidiah S. Snyder, Joshua V. Garn.

**Funding acquisition:** Matthew C. Freeman, Maryann G. Delea, Abebe Gebremariam Gobezayehu.

**Investigation:** Matthew C. Freeman, Maryann G. Delea, Jedidiah S. Snyder, Gloria D. Sclar, Yihenew Tesfaye, Mulat Woreta, Kassahun Zewudie, Abebe Gebremariam Gobezayehu.

**Methodology:** Matthew C. Freeman, Maryann G. Delea, Jedidiah S. Snyder, Joshua V. Garn, Bethany A. Caruso, Thomas F. Clasen.

**Project administration:** Matthew C. Freeman, Maryann G. Delea, Jedidiah S. Snyder, Mulusew Belew, Kassahun Zewudie, Abebe Gebremariam Gobezayehu.

**Supervision:** Matthew C. Freeman, Maryann G. Delea, Jedidiah S. Snyder, Mulusew Belew, Mulat Woreta, Kassahun Zewudie, Abebe Gebremariam Gobezayehu.

**Writing – original draft:** Matthew C. Freeman.

**Writing – review & editing:** Matthew C. Freeman, Maryann G. Delea, Jedidiah S. Snyder, Joshua V. Garn, Mulusew Belew, Bethany A. Caruso, Thomas F. Clasen, Gloria D. Sclar, Yihenew Tesfaye, Mulat Woreta, Kassahun Zewudie, Abebe Gebremariam Gobezayehu.

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
