## [Decision Letter · Decision Letter 0]

23 Sep 2021

PGPH-D-21-00373The impact of a demand-side sanitation and hygiene promotion intervention on sustained behavior change and health in Amhara, Ethiopia: a cluster-randomized trialPLOS Global Public Health

Dear Dr. Freeman

Thank you for submitting your manuscript to PLOS Global Public Health. After careful consideration, we feel that it has merit but does not fully meet PLOS Global Public Health’s publication criteria as it currently stands. Therefore, we invite you to submit a revised version of the manuscript that addresses the points raised during the review process.

We look forward to receiving your revised manuscript.

Kind regards,

Professor Audrey Prost

Academic Editor

Journal Requirements:

1. Please provide additional details regarding participant consent. In the ethics statement in the Methods and online submission information, please ensure that you have specified whether consent was informed.

2. If your study includes a novel survey or questionnaire, please ensure that you have provided sufficient details that others could replicate the analyses. For instance, if you developed a questionnaire as part of this study and it is not under a copyright more restrictive than CC-BY, please include a copy, in both the original language and English, as Supporting Information.

3. Please note that your Data Availability Statement is currently missing the repository name and a direct link to access each database. If your manuscript is accepted for publication, you will be asked to provide these details on a very short timeline. We therefore suggest that you provide this information now, though we will not hold up the peer review process if you are unable.

4.  We have noticed that you have uploaded supporting information but you have not included a list of legends.  Please add a full list of legends for all supporting information files (including figures, table and data files) after the references list.

Additional Editor Comments (if provided):

This is an important and well-conducted study. We recommend minor revisions. In addition to addressing the reviewers' comments, please also:

1. Review and correct the sentence on lines 112-3, which seems to be missing a clause.

2. Ensure that Table 1 is located in the results section, rather than the methods section.

Reviewers' comments:

Reviewer's Responses to Questions

**Comments to the Author**

1. Does this manuscript meet PLOS Global Public Health’s publication criteria? Is the manuscript technically sound, and do the data support the conclusions? The manuscript must describe methodologically and ethically rigorous research with conclusions that are appropriately drawn based on the data presented.

Reviewer #1: Yes

Reviewer #2: Yes

2. Has the statistical analysis been performed appropriately and rigorously?

Reviewer #1: Yes

Reviewer #2: Yes

3. Have the authors made all data underlying the findings in their manuscript fully available (please refer to the Data Availability Statement at the start of the manuscript PDF file)?

Reviewer #1: Yes

Reviewer #2: Yes

4. Is the manuscript presented in an intelligible fashion and written in standard English?

Reviewer #1: Yes

Reviewer #2: Yes

5. Review Comments to the Author

Reviewer #1: You have clearly conducted a scientifically sound analysis to investigate a contextually significant issue. I have a few minor suggestions for clarification. What is your baseline, and what distinguishes Andilaye from Community-Led Total Sanitation (CLTS), Participatory Hygiene and Sanitation Transformation (PHAST), and Ethiopian Health Extension Program at the start of its development? How to deal with the contamination of information on the during Andilaye intervention.

Reviewer #2: The authors do an excellent job motivating the study. I appreciated that this was an effectiveness study, contrasted very clearly in the discussion with an efficacy study to point out the challenges of scaling such interventions. I suggest some areas for improvement, but I felt that the paper overall was well written and concisely described the study and results. There are a couple instances where some field-specific terms could be better explained to improve accessibility for broader PLOS GPH readers. My main comment is for the discussion section, where I do think the low household exposure to activities should be further discussed. Overall, I commend the authors for this important study and clear reporting of results and would look forward to reading this in its final form in PLOS GPH.

Specific areas for improvement:

o I really do think the process evaluation results and particularly the low household exposure to the intervention should be included even more in the discussion. Are the null results surprising given the cascade of low implementation fidelity and even lower household exposure? Even if there is high dose delivered/received at levels above the household, outcomes are measured at the household level. I do very much appreciate the inclusion in the discussion of the problematic reliance on unpaid women’s labor (women as WDALs), but I would have liked to see additional discussion of the women respondents (especially as women’s mental well-being is addressed in detail in the intro as motivation for the study). The fact that there was an impact on reported sanitation-related social support, even with this low exposure, is promising. (And I’m noting also the non-statistically-significant, but still trending toward improved, mental health outcomes shown in Table 2.)

Minor issues

o Missing space in abstract

o Harmonize format of reporting PRs in abstract with colon or equal sign or space

o Some very minor copy-editing errors throughout

o Table 1:

I strongly suggest reducing the number of significant figures for the PDs to two decimal places

It would be helpful to include the total sample size (N) for each of the intervention and control groups at the top of the column, in addition to the number with each outcome (n) that is currently included

o Line 155: Suggest removing “clusters” and just referring to kebeles as the randomization units (“clusters” made me wonder if the unit was multiple kebeles)

o Line 157: It is not totally clear how topographical conditions are related to the intervention, though I can speculate. I suggest editing to make that reasoning and link more clear.

o Line 188 onward: It’s not clear to me from the text how the intervention activities are delivered at the district level, if the randomization occurred at the kebele level (so there are both intervention and control groups within the district). I suggest clarifying a bit.

o Line 193: Could you add some explanation of the ‘Whole System in the Room’ activity as well as of the cross-fertilization visits? The former comes up multiple times throughout the paper but is not clearly described.

o General note in intervention explanation – “district” and “household” seem well defined, but “group” and “community” are not. I think it would be helpful to explain a bit what is meant by those two levels.

o Line 216 and 268: “Ethiopia-based”

o Line 219: “standard of care”

o Line 220: Are these equivalence tests described anywhere? Looking at Figure 1, I agree that 19/25 and 20/25 are balanced arms, but I’m not sure that should be described as an equivalence analysis.

o Line 221: “Baseline statistics showing X, along with…” (suggest expanding for clarity)

o Line 249: For diarrhea period prevalence, can you specify (e.g., 7-day and 2-day) recall period?

o Line 284: Does this percent range include other disorders as well? Or specifically anxiety and depression? If the latter, please delete “of common mental disorders”

o Under analytical methods, please include which software/estimation commands that were used to estimate marginal effects

o Line 307: “accounted”

o Line 328: “…and well exceeded our minimum sample size requirement…”

o Lin 331: suggest replacing “by design” with “who were typically the primary caretakers”, or something similar

o Overall, I think it’s helpful for the reader when they are provided some quantitative metric alongside general descriptions of data such as “conditions were often poor” (e.g., line 380), especially if the evidence supporting this will be in SI rather than main text. For example, in this line, you could move the “e.g., presence of feces” to later in the sentence and also include the accompanying proportions.

o Table 2 & Table 3:

It seems like the differences should use fewer significant figures (the scores themselves only go to two decimal places)

o Table 3 caption – delete last “scale”

o Line 450: just confirming all of the reported PRs are adjusted

o Line 513-517: Thank you for including this very important point!

o Line 541: Please add a brief explanation of the “fried egg” approach.

o Line 562-564: Great takeaway and so important

o I think I am missing the full Figure 3 – unsure what the x-axis labels are

o Figure 1: What’s a gott?

o SI tables: Check significant figures

o What is the WDAL exactly? This is unclear from the main text, but would be helpful to clarify, especially because the point about increasing responsibilities asked of WDALs is brought up as a problem.

6. PLOS authors have the option to publish the peer review history of their article (what does this mean?). If published, this will include your full peer review and any attached files.

**Do you want your identity to be public for this peer review?** For information about this choice, including consent withdrawal, please see our Privacy Policy.

Reviewer #1: No

Reviewer #2: No

---

## [Decision Letter · Decision Letter 1]

22 Nov 2021

PGPH-D-21-00373R1

The impact of a demand-side sanitation and hygiene promotion intervention on sustained behavior change and health in Amhara, Ethiopia: a cluster-randomized trial

Dear Dr. Freeman,

Thank you for submitting your manuscript to PLOS Global Public Health. After careful consideration, we feel that it has merit but does not fully meet PLOS Global Public Health’s publication criteria as it currently stands. Therefore, we invite you to submit a revised version of the manuscript that addresses the points raised during the review process.

We look forward to receiving your revised manuscript.

Kind regards,

Reginald Quansah, Ph.D.

Academic Editor

Journal Requirements:

Additional Editor Comments (if provided):

Reviewers' comments:

Reviewer's Responses to Questions

**Comments to the Author**

1. If the authors have adequately addressed your comments raised in a previous round of review and you feel that this manuscript is now acceptable for publication, you may indicate that here to bypass the “Comments to the Author” section, enter your conflict of interest statement in the “Confidential to Editor” section, and submit your "Accept" recommendation.

Reviewer #1: All comments have been addressed

Reviewer #2: All comments have been addressed

2. Does this manuscript meet PLOS Global Public Health’s publication criteria? Is the manuscript technically sound, and do the data support the conclusions? The manuscript must describe methodologically and ethically rigorous research with conclusions that are appropriately drawn based on the data presented.

Reviewer #1: Yes

Reviewer #2: Yes

3. Has the statistical analysis been performed appropriately and rigorously?

Reviewer #1: Yes

Reviewer #2: Yes

4. Have the authors made all data underlying the findings in their manuscript fully available (please refer to the Data Availability Statement at the start of the manuscript PDF file)?

Reviewer #1: Yes

Reviewer #2: Yes

5. Is the manuscript presented in an intelligible fashion and written in standard English?

Reviewer #1: Yes

Reviewer #2: Yes

6. Review Comments to the Author

Reviewer #1: Thank you for taking the time to respond to each of my comments individually.

Reviewer #2: The authors have done an excellent job responding to my comments. The revisions have greatly improved the clarity of the paper, and I have no remaining substantive concerns. This will be a valuable contribution to the literature.

I have 3 minor copy-editing suggestions (line numbers in the track-changed doc) in the edited lines:

line 216 - "dialogue"

line 578 - remove comma

line 584 - "non-statistically significant"

7. PLOS authors have the option to publish the peer review history of their article (what does this mean?). If published, this will include your full peer review and any attached files.

**Do you want your identity to be public for this peer review?** For information about this choice, including consent withdrawal, please see our Privacy Policy.

Reviewer #1: **Yes: **Negasa Eshete Soboksa

Reviewer #2: No

---

## [Editor Report · Decision Letter 2]

7 Dec 2021

The impact of a demand-side sanitation and hygiene promotion intervention on sustained behavior change and health in Amhara, Ethiopia: a cluster-randomized trial

PGPH-D-21-00373R2

Dear Dr. Freeman,

We're pleased to inform you that your manuscript has been judged scientifically suitable for publication and will be formally accepted for publication once it meets all outstanding technical requirements.

Within one week, you'll receive an e-mail detailing the required amendments. When these have been addressed, you'll receive a formal acceptance letter and your manuscript will be scheduled for publication.

An invoice for payment will follow shortly after the formal acceptance. To ensure an efficient process, please log into Editorial Manager at https://www.editorialmanager.com/pgph/ click the 'Update My Information' link at the top of the page, and double check that your user information is up-to-date. If you have any billing related questions, please contact our Author Billing department directly at authorbilling@plos.org.

Kind regards,

Reginald Quansah, Ph.D.

Academic Editor